# XBB.1.5 RBD-Based Bivalent Vaccines Induced Antibody Responses Against SARS-CoV-2 Variants in Mice

**DOI:** 10.3390/vaccines13050543

**Published:** 2025-05-20

**Authors:** Jiawen Liu, Tiantian Wang, Hongying Ren, Ruixi Liu, Qian Wang, Jun Wu, Bo Liu

**Affiliations:** 1Laboratory of Advanced Biotechnology, Beijing Institute of Biotechnology, Beijing 100071, China; a807399505@outlook.com (J.L.); con_an@126.com (T.W.);; 2Key Laboratory of Animal Genetics, Breeding and Reproduction of Shaanxi Province, College of Animal Science and Technology, Northwest A&F University, Yangling 712100, China

**Keywords:** SARS-CoV-2, RBD, subunit vaccine, monovalent vaccine, bivalent vaccine, glycoengineered yeast

## Abstract

(1) Background: The currently circulating variant of severe acute respiratory syndrome coronavirus 2 (SARS-CoV-2) exhibits resistance to antibodies induced by vaccines. The World Health Organization recommended the use of monovalent XBB.1 sublineages (e.g., XBB.1.5) as an antigenic component in 2023. (2) Objective: In this study, we aimed to develop vaccines based on the XBB.1.5 receptor-binding domain (RBD) to combat the recently emerged SARS-CoV-2 XBB and JN.1 variants, as well as previously circulating variants. (3) Methods: Glycoengineered *Pichia pastoris* was utilized to produce a recombinant XBB.1.5 RBD protein with mammalian-like and fucose-free N-glycosylation. The XBB.1.5 RBD was mixed with Al(OH)_3_:CpG adjuvants to prepare monovalent vaccines. Thereafter, the XBB.1.5 RBD was mixed with the Beta (B.1.351), Delta (B.1.617.2), or Omicron (BA.2) RBDs (1:1 ratio), along with Al(OH)_3_:CpG, to prepare bivalent vaccines. BALB/c mice were immunized with the monovalent and bivalent vaccines. Neutralizing antibody titers were assessed via pseudovirus and authentic virus assays; humoral immune responses were analyzed by RBD-binding IgG subtypes. (4) Results: The monovalent vaccine induced higher neutralizing antibody titers against Delta, BA.2, XBB.1.5, and JN.1 compared to those in mice immunized solely with Al(OH)_3_:CpG, as demonstrated by pseudovirus virus assays. The XBB.1.5/Delta RBD and XBB.1.5/Beta RBD-based bivalent vaccines provided potent protection against the BA.2, XBB.1.5, JN.1, and KP.2 variants, as well as the previously circulating Delta and Beta variants. All monovalent and bivalent vaccines induced high levels of RBD-binding IgG (IgG1, IgG2a, IgG2b, and IgG3) antibodies in mice, suggesting that they elicited robust humoral immune responses. The serum samples from mice immunized with the XBB.1.5 RBD-based and XBB.1.5/Delta RBD-based vaccines could neutralize the authentic XBB.1.16 virus. (5) Conclusions: The XBB.1.5/Beta and XBB.1.5/Delta RBD-based bivalent vaccines are considered as potential candidates for broad-spectrum vaccines against SARS-CoV-2 variants.

## 1. Introduction

Severe acute respiratory syndrome coronavirus 2 (SARS-CoV-2), which emerged in 2019, caused the COVID-19 pandemic, creating an urgent demand for vaccination [1]. This virus is an enveloped, single-stranded, positive-sense RNA virus [2] composed of a nucleocapsid, membrane, envelope, and glycosylated spike proteins [3]. The spike proteins, which are abundant on the virion surface, are anchored to the viral membrane and mediate the fusion between viral and host cell membranes [4,5]. The spike protein (1273 amino acids [aa] long) is cleaved into S1 and S2 subunits by a furin-like protease during membrane fusion and viral entry [4]. The S1 subunit contains a receptor-binding domain (RBD; 319–541 aa) and an amino-terminal (N-terminal) domain (NTD) [4]. The RBD domain binds to human angiotensin-converting enzyme 2 (ACE2) [6], making it an attractive vaccine target for the blocking of receptor binding [7,8].

RNA viruses often mutate during transmission, resulting in the emergence of new variants that can trigger pandemics [9,10,11]. Mutated SARS-CoV-2 strains have a higher transmission rate than the original variants and have been termed “variants of concern” (VOCs) [12]. Several VOCs emerged independently and became dominant at both the local and global scales by replacing earlier variants, including the VOCs Alpha (B.1.1.7), Beta (B.1.351), Gamma (P.1), Delta (B.1.617.2), and Omicron (B.1.1.529) (https://www.who.int/publications/m/item/updated-working-definitions-and-primary-actions-for--sars-cov-2-variants; accessed on 4 October 2023). The Omicron variant has evolved into several sublineages, including BA.2.75, BA.4/5, BF.7, XBB, BA.2.86, and JN.1; among these, the XBB.1.5 subvariant from the XBB lineage is a recombinant of two BA.2 lineages with an additional RBD mutation (S486P), which enhances its receptor binding [13,14,15]. XBB.1.5 is one of the most antibody-resistant variants to date due to decreased antibody-mediated immunity against the Omicron XBB subvariant, suggesting a need for an updated vaccine [16,17]. In addition to the XBB lineage, BA 2.86 subvariants such as JN.1 and KP.2 have come to prominence [18,19]. The World Health Organization (WHO; https://www.who.int/news/item/18-05-2023-statement-on-the-antigen-composition-of-covid-19-vaccines; accessed on 18 May 2023), European Medicines Agency (EMA; https://www.ema.europa.eu/en/news/ema-ecdc-statement-updating-covid-19-vaccines-target-new-sars-cov-2-virus-variants; accessed on 6 June 2023), and United States Food and Drug Administration (FDA; https://www.fda.gov/media/169591/download; accessed on 11 September 2023) recommended the use of the Omicron XBB.1.5-based vaccine to protect against the 2023–2024 XBB sublineage. Consequently, several mRNA- and protein-based vaccine manufacturers, including Moderna, Pfizer-BioNTech, and Novavax, updated the antigenic composition of the COVID-19 vaccine to obtain monovalent XBB.1.5 formulations (https://www.fda.gov/news-events/press-announcements/fda-authorizes-updated-novavax-covid-19-vaccine-formulated-better-protect-against-currently; accessed on 3 October 2023).

The SARS-CoV-2 spike protein comprises 22 N-glycosylation sites; of these, two sites (N331 and N343) located within the RBD play an important role in protein folding and immune evasion [7,8,9,10]. The deletion of both N331 and N343 glycosylation drastically reduces SARS-CoV-2’s infectivity and the RBD’s binding affinity to ACE2 [11,12]. The glycosylation of the RBD is associated with antigen presentation, humoral immunogenicity, and immunoreactivity [7]. An RBD with N-glycosylation is considered as a potential target for vaccine development. The *Pichia pastoris* expression system is widely used for its low cost and its ability to produce heterologous proteins in the desired yield and quality [20]. However, the high-mannose proteins expressed in yeast may lead to non-specific immune recognition [21]. The RBD proteins with mammalian N-glycosylation modifications were expressed in glycoengineered *P. pastoris* in our lab [22,23,24,25].

Adjuvants assist in boosting specific immune responses to the antigens found in vaccines [26]. CpG2006 (CpG-7909), a synthetic oligodeoxynucleotide containing CpG motifs (‘CpG ODN’), belongs to the class of TLR9 agonists, which can be recognized by Toll-like receptor 9 (TLR9) [27]. A CpG effectively elicited the secretion of Th1-type cytokines, including interleukin (IL)-6, IL-12, and interferon gamma, through detection by TLR9 in B, T, and natural killer cells [28]. Aluminum hydroxide [Al(OH)_3_] is the most commonly used aluminum-based adjuvant, preferentially stimulating Th2-type immune responses by enhancing antigen persistence and recruiting innate immune cells [29]. In previous studies, bivalent RBD vaccines adjuvanted with 50 µg CpG and 100 µg Al(OH)_3_ have been shown to induce a robust antibody response in mice and provide broad-spectrum protection [24,25]. In addition to protecting against the currently dominant variants, bivalent vaccines can also neutralize previously dominant and potential future variants [30,31,32]. The Omicron/Delta RBD-based bivalent vaccines still confer protection against Omicron subvariants; however, their antibody-mediated abilities against the recently emerged Omicron variants, such as BA.4 and BA.5, are reduced. The neutralizing ability of the Omicron/Delta RBD-based bivalent vaccines against recently emerged variants such as the XBB and JN.1 sublineages may also be reduced. In response to the prevalence of the XBB and JN.1 subvariants, we designed XBB.1.5 RBD-based monovalent and bivalent vaccines and assessed their neutralizing effects against the recently emerged SARS-CoV-2 XBB and JN.1 variants and previously circulating variants.

## 2. Materials and Methods

### 2.1. Ethics Statement

This study was approved by the Animal Center of the Beijing Institute of Biotechnology (welfare ethics number: IACUC-2023-025).

### 2.2. Experimental Models and Reagents

*Escherichia coli* Trans5α was purchased from TransGen Biotech (Beijing, China); glycoengineered *P. pastoris* was constructed in our lab [33]; and BALB/c mice were purchased from the Weitonglihua Experimental Animal Factory (Beijing, China). The Beta, Delta, and BA.2 RBD protein samples were prepared and stored in our laboratory [24,32]. The pPICZαA vector was purchased from Invitrogen (Carlsbad, CA, USA). Tryptone and yeast extract were purchased from OXOID (Thermo Fisher Scientific, Waltham, MA, USA); glucose was obtained from Shanghai Chemical Reagent Co., Ltd. (Shanghai, China); and hygromycin B, zeocin, and G418 were obtained from Thermo Fisher Scientific (Waltham, MA, USA). The anti-SARS-CoV-2 spike S1 antibody (rabbit) was produced and kept in our laboratory.

### 2.3. XBB.1.5 RBD Protein Expression, Fermentation, and Purification

Table 1 shows mutations in the RBDs of SARS-CoV-2 variants Beta, Delta, BA.2, and XBB.1.5 relative to the prototype. The SRAS-CoV-2 *XBB.1.5 RBD* gene (OQ054680.1, GenBank) was synthesized through a polymerase chain reaction (PCR) and ligated into the pPICZαA vector at the *Xho*I/*Not*I sites to construct a recombinant pPICZαA-*XBB.1.5 RBD* vector. The plasmid was linearized by *Bg1*II and transformed into *P. pastoris* by electroporation. The transformed plasmids were recognized by colony PCR and DNA sequencing analysis using *alcohol oxidase* primers (forward: 5′-GACTGGTTCCAATTGACAAGC-3′ and reverse: 5′-GGCAAATGGCATTCTGACAT-3′).

The recombinant RBD protein expression and purification protocols were previously reported [24]. The transformed *P. pastoris* cells were cultured in a shake flask and treated daily with methanol (1% of the total volume) to induce protein expression. After 3 d of induction, the supernatants were collected and the recombinant C-terminal His-tagged XBB.1.5 RBD protein was isolated by nickel affinity chromatography. A high-quality RBD protein was obtained through protein dialysis, ultrafiltration, and gel affinity chromatography (Superdex-G75; GE Healthcare, Anaheim, CA, USA).

### 2.4. Recombinant RBD Protein Characterization

The recombinant XBB1.5, Beta, Delta, and BA.2 RBD proteins were digested with Peptide-N-Glycosidase F (PNGase F) and analyzed through sodium dodecyl sulfate–polyacrylamide gel electrophoresis (SDS-PAGE) and Western blot using rabbit anti-SARS-CoV spike S1 antibody and horseradish peroxidase (HRP)-conjugated goat anti-rabbit IgG antibodies (1:3000; Sigma-Aldrich, St. Louis, MO, USA). The purity of the RBD was analyzed by size-exclusion high-performance liquid chromatography (SEC-HPLC) through an Agilent Technologies 1260 Infinity HPLC (Agilent Technologies Inc., Santa Clara, CA, USA), and the absorbance of the obtained protein samples was recorded at 280 nm, as previously described [24].

The molecular mass of the XBB.1.5 RBD was assessed. The RBD proteins were digested with PNGase F. Subsequently, the protein samples were separated using ultra-performance liquid chromatography (UPLC) (Acquity UPLC I-Class; Waters, Milford, MA, USA). The protein’s molecular mass was then determined by mass spectrometry (MS) (XevoG2-XS QTOF; Waters).

Glycosyl groups were evaluated through MS and UPLC. Proteins were transferred to phosphate buffer (PB) by ultrafiltration and then were incubated with sodium dodecyl sulfate (SDS) at room temperature. The sugar chains on the RBD glycoproteins were excised using PNGase F. After incubation with ethanol, the supernatant containing oligosaccharides was isolated by centrifugation and was freeze-dried in a vacuum. Oligosaccharides were labeled with 2-aminobenzamide (2-AB; Sigma-Aldrich, St. Louis, MO, USA) for fluorescent detection. The 2-AB-labeled oligosaccharides were isolated via the Waters Acquity UPLC I-Class liquid phase system and then confirmed and analyzed via the XevoG2-XS QTOF (Waters).

### 2.5. In Vivo Analysis

Female mice, aged 7 weeks, were randomly assigned to 10 groups (10 mice/group) and intramuscularly injected (100 μL) with the vaccines on days 0 and 14. On day 28, the mice were subjected to retro-orbital blood sampling (Figure 1A).

The mouse immunization regimens were as follows (Figure 1B): 5 µg XBB.1.5 RBD and 5 µg Beta RBD (Group 1); 5 µg XBB.1.5 RBD and 5 µg BA.2 RBD (Group 2); 5 µg XBB.1.5 RBD and 5 µg Delta RBD (Group 3); 10 µg XBB.1.5 RBD (Group 4); 5 µg XBB.1.5 RBD (Group 5); 2.5 µg XBB.1.5 RBD (Group 6); 1.25 µg XBB.1.5 RBD and 1.25 µg Beta RBD (Group 7); 1.25 µg XBB.1.5 RBD and 1.25 µg BA.2 RBD (Group 8); and 1.25 µg XBB.1.5 RBD and 1.25 µg Delta RBD (Group 9). In all immunization programs, the RBD proteins were mixed with two adjuvants—100 µg aluminum hydroxide [Al(OH)_3_] (CRODA, Elsenbakken, Denmark) and 50 µg CpG2006 (TGCTCGTTTTGTGCTTTTGTGCTT). Meanwhile, the control group (Group 10) was injected with 100 µg [Al(OH)_3_]–50 µg CpG adjuvant. Groups 1–4 were high-dose groups, Group 5 was a medium-dose group, and Groups 6–9 were low-dose groups.

### 2.6. Enzyme-Linked Immunosorbent Assay (ELISA)

The antibody titers of serum samples from the high-dose groups (Group 1–4) and control group (Group 10) were tested through ELISA. After 6–8 h of incubation at 4 °C, the blood samples were centrifuged at 8000 rpm for 20 min, and the supernatant (containing serum) was aspirated. Sera from the control and test groups were diluted in 1:50 and 1:5000 ratios, respectively, with phosphate-buffered saline (PBS) with 0.1% Tween 20 (PBST) supplemented with 5% skim milk. The serum samples were then serially diluted 3-fold. The control group sera, diluted in a 1:1000 ratio, were used as blanks.

A 96-well plate was treated with 2 µg/mL of XBB.1.5-specific RBD protein (in PBS) and incubated at 4 °C overnight. The plates were washed thrice with PBST and blocked for 1 h with 5% skim milk in PBST (300 µL/well) at 37 °C. Subsequently, serum samples were diluted and added at this stage and incubated for 1 h 37 °C. Then, 100 µL of anti-mouse IgG HRP-conjugated antibody (1:3000; Sigma-Aldrich, St. Louis, MO, USA) was added to each well. After 1 h of incubation at 37 °C, the plates were washed four times with PBST and treated with TMB one-component chromogenic solution (100 μL/well) for 3 min. The reaction was terminated by adding 2 M H_2_SO_4_ (50 μL/well). Absorbance was measured at 450 and 630 nm.

### 2.7. Pseudovirus Neutralization Assay

The neutralizing antibody titers of the serum samples of the mice from Groups 1–10 were evaluated through a pseudovirus neutralization assay. The serum from Groups 1–5 was the same as that utilized in the ELISA. ACE2/293T cells were added to a 96-well plate and cultured overnight at 37 °C and 5% CO_2_. Serum samples were heated for 30 min at 56 °C and diluted (1:30 or 1:50) with Dulbecco’s Modified Eagle Medium (DMEM; Gibco, Grand Island, NY, USA) containing 10% fetal bovine serum (FBS; Excell, Suzhou, China) and 1% penicillin–streptomycin (WISENT, Nanjing, China) in a 1:30 or 1:50 ratio and then serially diluted. The SARS-CoV-2-Fluc pseudovirus variants (Beta, Delta, BA.2, XBB.1.5, KP.2, and JN.1) were incubated with diluted sera for 1 h at 37 °C. Thereafter, the sera and pseudovirus mixture were added to the ACE2/293T cells. After 8 h of incubation, the cell culture supernatant was aspirated, fresh DMEM was supplemented with 10% FBS and 1% penicillin–streptomycin was added to each well, and the cells were cultured for 48 h. Finally, 100 µL of the cell culture supernatant was removed, and 100 µL of luciferase assay substrate (Vazyme, Nanjing, China) was added to each well. After 2 min of shaking and 5 min of incubation, the samples were analyzed using a fluorescent enzyme labeler. The median effective dose 50% (ED_50_) values were analyzed and visualized using GraphPad Prism 9.5.

### 2.8. Authentic Virus Neutralization Assay

These experiments were conducted at Sinopharm Zhongsheng Biotechnology Research Institute Co. (Beijing, China). We evaluated the neutralizing antibody titers against XBB.1.16 of the serum samples from Groups 3, 4, and 10, i.e., the 5 µg XBB.1.5/5 µg Delta RBD immunization group, the 10 µg XBB.1.5 RBD immunization group, and the control group. Vero cells were used to assess the neutralizing antibody titers. Serum samples were diluted from 1:10 to 1:20,480 in a 2-fold series, while the XBB.1.16 virus was diluted to 100 TCID_50_/50 µL. The diluted viral and serum samples were mixed and incubated with the Vero cells at 37 °C and 5% CO_2_ for 4 d. The results were measured as described above.

### 2.9. Statistical Analysis

Data were processed using GraphPad Prism 9.5 and statistically analyzed via a *t*-test. Statistical significance is indicated by * *p* < 0.05; ** *p* < 0.01; *** *p* < 0.001; and **** *p* < 0.0001, and statistical non-significance is indicated by ‘ns’.

## 3. Results

### 3.1. The Recombinant XBB.1.5 RBD Protein Was Produced in Glycoengineered P. pastoris

Figure 2A shows the position of the RBD on the SARS-CoV-2 spike protein and the mutations in the RBD proteins of the Beta, Delta, BA.2, and XBB.1.5 variants. The *XBB.1.5 RBD* gene was cloned into the pPICZαA plasmid and expressed through glycoengineered *P. pastoris*. The SDS-PAGE and Western blot analyses revealed that the molecular mass of all PNGaseF-treated RBD proteins decreased by approximately 5 kDa, indicating that the expressed RBD proteins were N-glycosylated (Figure 2B). The obtained XBB.1.5, Beta, Delta, and BA.2 RBD proteins were found to be >97% pure, as indicated by SEC-HPLC analysis (Figure 2C). Furthermore, high-resolution XevoG2-XS QTOF MS analysis showed that the molecular mass of the deglycosylated XBB.1.5 RBD protein was 25.127 kDa, which was almost consistent with its theoretical mass of 25.123 kDa (Figure 2D). Then, the N-glycans in the XBB.1.5 RBD were identified and quantified, and the results revealed that the XBB.1.5 RBD predominantly contained mammalian A2G(4)2 glycan (Figure 2E).

### 3.2. XBB.1.5 RBD-Based Monovalent and Bivalent Vaccines Induced a Robust Humoral Immune Response in Mice

To investigate the immune responses of mice to different combinations of RBD vaccines, BALB/c mice were immunized on days 0 and 14 with XBB.1.5 RBD-based monovalent or bivalent vaccines, and serum was collected at 14 d after the second immunization (Figure 3A). The ELISA results showed that the XBB.1.5/Beta RBD, XBB.1.5/BA.2, XBB.1.5/Delta RBD, and XBB.1.5 RBD (Groups 1–4) induced high IgG titers (1.09 × 10^7^, 9.80 × 10^6^, 7.86 × 10^6^, and 2.60 × 10^7^, respectively) against the SARS-CoV-2-specific RBD in mice (Figure 3B). Further analysis revealed a significant increase (*p* < 0.0001) in different IgG subtypes (IgG1, IgG2a, IgG2b, and IgG3) in Groups 1–4 (Figure 3C–F) compared with that of Group 10 (control group). The ratio of IgG1 to IgG2a was approximately 1, indicating a balanced Th1/Th2 immune response in the mice, consistent with our previous study [25]. Additionally, Groups 1–3 (bivalent) and Group 4 (monovalent) showed no significant differences (*p* > 0.05) in the levels of IgG and its subtypes. Therefore, the XBB.1.5 RBD-based monovalent and bivalent vaccines induced robust humoral immune responses in mice.

### 3.3. XBB.1.5 RBD-Based Monovalent Vaccine Induced High Neutralizing Antibody Titers Against Delta, BA.2, XBB.1.5, JN.1, and KP.2 VOCs, but Not Beta

To assess the efficacy of the XBB.1.5 RBD-based monovalent vaccines, the mice in Groups 4, 5, and 6 were immunized with 10, 5, or 2.5 µg of the recombinant XBB.1.5 RBD, respectively (Figure 1B). Mouse sera were collected at 14 d after the second booster to test against SARS-CoV-2 pseudoviruses (Figure 4), including early Beta, Delta, BA.2, and XBB.1.5 variants, as well as the currently prevalent JN.1 and KP.2 variants from the BA.2.86 lineage (Figure 4).

For the neutralizing antibodies against the SARS-CoV-2 XBB.1.5 pseudovirus, the geometric mean of the ED_50_ of mice vaccinated with 10 µg (Group 4), 5 µg (Group 5), or 2.5 µg (Group 6) XBB.1.5 RBD was 1:324, 1:676, and 1:372, respectively, compared to that of the control group serum (Figure 5A). Sera from Groups 4, 5, and 6 showed no significant neutralizing effects against the Beta pseudovirus, suggesting that the XBB.1.5 RBD fails to produce neutralizing antibodies against the Beta variant (Figure 5B). For the neutralizing antibodies against the Delta pseudovirus, the geometric mean of the ED_50_ of the sera from Groups 4, 5, and 6 reached 1:955, 1:2555, and 1:1122, respectively (Figure 5C). For the neutralizing antibodies against the BA.2 pseudovirus, the geometric mean of the ED_50_ of the sera from Groups 4, 5, and 6 reached 1:355, 1:501, and 1:219 (*p* < 0.0001), respectively (Figure 5D). For the neutralizing antibodies against the JN.1 pseudovirus, the geometric means of the ED_50_ of the sera from Groups 4, 5, and 6 were 1:398, 1:1175, and 1:813. However, there were no significant differences in the neutralizing capacity of the high-/medium-/low-dose XBB.1.5 RBD immunization groups. The geometric means of the ED_50_ of the sera from Group 4 (10 µg XBB.1.5 RBD immunization) and Group 5 (5 µg XBB.1.5 RBD immunization) reached 1:234 and 1:886. However, the serum from Group 6 (2.5 µg XBB.1.5 RBD immunization) exhibited a profound loss of neutralizing activity against the KP.2 pseudovirus, with the geometric mean of the ED_50_ values dropping to 1:81. These results indicate that monovalent vaccines composed of 10 or 5 µg of the XBB.1.5 RBD can induce high neutralizing antibody titers against the SARS-CoV-2 Delta, BA.2, XBB.1.5, JN.1, and KP.2 variants.

To further validate the neutralizing effects of sera from immunized mice against authentic VOCs, we analyzed the sera against the SARS-CoV-2 XBB.1.16 variant. XBB.1.16, a descendant of XBB.1.5, is relatively stronger than XBB.1.5 in terms of transmissibility and immune escape, which is mainly attributed to its E180V and K478R mutations [34]. Sera from the 10 µg XBB.1.5 RBD-immunized group (Group 4) exhibited a robust neutralizing capacity against the XBB.1.16 virus (*p* < 0.0001), with the geometric mean of the ED_50_ value being 1:1148, consistent with the pseudovirus neutralization results (Figure 5G). Thus, these monovalent vaccines comply with the guidelines provided by the WHO, FDA, and EMA for updated vaccine compositions during 2023–2024.

### 3.4. The XBB.1.5/BA.2 RBD-Based Bivalent Vaccine Did Not Enhance the Neutralizing Response or Cross-Protection Compared with Monovalent Vaccines

We evaluated the neutralizing effects of sera from mice immunized with two doses of 5 μg XBB.1.5/5 μg BA.2 RBD (Group 2) and 1.25 μg XBB.1.5/1.25 μg BA.2 RBD (Group 8). The 5 μg XBB.1.5/5 μg BA.2 RBD bivalent vaccine induced a neutralizing effect against the XBB.1.5, Delta, BA.2, JN.1, and KP.2 VOCs, with the geometric mean of the ED_50_ values of 1:186, 1:851, 1:166, 1:631, and 1:151, respectively (Figure 5A,C–F). However, the neutralizing activity of the sera against SARS-CoV-2 variants from the XBB.1.5/BA.2 RBD-immunized group was not significantly higher than that of the sera from the XBB.1.5 RBD-immunized group (Group 4). Similarly to the XBB.1.5 RBD-based monovalent vaccine (Figure 5A,C–E), the XBB.1.5/BA.2 RBD-based bivalent vaccine failed to induce neutralizing antibodies against Beta in mice. Previous studies have shown that serum from mice immunized with the BA.1/BA.2 vaccine is ineffective in neutralizing Beta pseudoviruses [12]. In addition, the low-dose XBB.1.5 RBD-immunized group only exhibited a potent neutralizing response against Delta (*p* < 0.01) and not against XBB.1.5, Beta, BA.2, JN.1, and KP.2. These results indicate that, compared with the XBB.1.5 RBD-based monovalent vaccine, the XBB.1.5/BA.2-based bivalent vaccine does not show a significantly enhanced neutralizing response or cross-protection in mice against SARS-CoV-2 variants.

### 3.5. The XBB.1.5/Beta RBD and XBB.1.5/Delta RBD-Based Bivalent Vaccines Enhanced the Broad-Spectrum Protection Against SARS-CoV-2 VOCs, Including Beta

We further evaluated the efficacy of other XBB.1.5 RBD-based bivalent vaccines, including the XBB.1.5/Beta RBD and XBB.1.5/Delta RBD vaccines, through a pseudovirus neutralization test. The high-dose XBB.1.5/Beta RBD immunization group (Group 1) showed neutralizing abilities against XBB.1.5, Beta, Delta, BA.2, JN.1, and KP.2, with the geometric mean of the ED_50_ of 1:209, 1:1318, 1:776, 1:209, 1:257, and 1:234, respectively (Figure 5A–F). Notably, the high-dose XBB.1.5/Beta RBD induced robust neutralizing antibodies against Beta in mice. In contrast, sera from the low-dose XBB.1.5/Beta RBD immunization group (Group 7) elicited higher neutralizing antibody titers against JN.1 (*p* < 0.001) compared with that of the control group but was ineffective in generating neutralizing antibodies against XBB.1.5, Beta, Delta, BA.2, and KP.2. The high-dose XBB.1.5/Delta RBD immunization group (Group 3) exhibited neutralizing activity against XBB.1.5, Beta, Delta, BA.2, JN.1, and KP.2. with the geometric mean of the ED_50_ values of 1:692, 1:166, 1:2239, 1:562, and 1:851, respectively (Figure 5A–F). Notably, sera from the low-dose XBB.1.5/Delta RBD immunization group (Group 9) showed neutralizing activity against XBB.1.5 (*p* < 0.05), Beta (*p* < 0.001), Delta (*p* < 0.001), and JN.1 (*p* < 0.05) compared with that of the control group. However, the neutralizing effects of the low-dose and high-dose XBB.1.5/Delta RBD immunization groups were not significantly different. Similarly to the monovalent vaccine-immunized group (Group 6), mouse antibodies stimulated by the low-dose XBB.1.5/Delta RBD were ineffective against the KP.2 pseudovirus (Figure 5F). These results indicate that the high-dose XBB.1.5/Beta and XBB.1.5/Delta RBD vaccines induce potent neutralizing antibody titers against SARS-CoV-2 variants Beta, Delta, BA.2, XBB.1.5, JN.1, and KP.2.

In addition, we evaluated the neutralizing antibodies obtained from the 5 µg XBB.1.5/5 µg Delta RBD immunization group (Group 3) against the SARS-CoV-2 variant XBB.1.16. The results revealed that the high-dose XBB.1.5/Delta RBD-based bivalent vaccine induced a strong neutralizing antibody response against the XBB.1.16 variant (*p* < 0.0001), consistent with the pseudovirus neutralization results (Figure 5G). Moreover, there was no significant difference in the neutralizing antibody titers of the XBB.1.5 and XBB.1.5/Delta RBD immunization groups against XBB.1.16. Therefore, the XBB.1.5/Beta RBD and XBB.1.5 Delta RBD-based bivalent vaccines are promising broad-spectrum vaccines for SARS-CoV-2 variants, especially compared with the XBB.1.5 RBD-based monovalent vaccine.

## 4. Discussion

The high mutation rate of the SARS-CoV-2 genome has led to the continuous generation of variants, with varying pathogenicity, immunological resistance, and transmissibility [35]. However, the efficacy of the vaccine against emerging mutant strains is diminished by mutations in the spike protein, particularly in the RBD, which is a target of the existing SARS-CoV-2 vaccines [35,36]. Our lab previously developed bivalent vaccines including prototype/Beta RBD and Omicron/Delta RBD adjuvanted with CpG and Al(OH)_3_, which can induce a robust antibody response in mice and provide broad-spectrum protection against Beta, Delta, and Omicron sublineages. However, they are less effective against the Omicron BA.4 and BA.5 subvariants compared to earlier SARS-CoV-2 variants, although they still provide protection against Omicron subvariants. Vaccines produced with proteins from new SARS-CoV-2 variants only provide transient protection against future variants [37]. Therefore, there is an urgent need to develop universal vaccines that are more effective in producing neutralizing antibodies against novel SARS-CoV-2 variants with enhanced immune escape abilities. Multivalent vaccines based on key mutations in the SARS-CoV-2 spike protein elicit broad protection against SARS-CoV-2 variants [24,25,38]. In this study, we developed XBB.1.5 RBD-based monovalent vaccines formulated with an Al(OH)_3_:CpG adjuvant that induced neutralizing effects against te=he Delta, BA.2, XBB.1.5, XBB.1.16, JN.1, and KP.2 variants but showed low effectiveness against the KP.2 variant. The XBB.1.5/Beta RBD and XBB.1.5/Delta RBD-based bivalent vaccines enhanced the cross-protection against SARS-CoV-2 variants including Beta compared with XBB.1.5 RBD-based monovalent vaccines.

The antibodies induced by the XBB.1.5 RBD-based monovalent vaccine effectively neutralized the XBB subvariants XBB.1.16, XBB.1.5, and JN.1 and the previously circulating BA.2 and Delta. In addition, immunization with high- or medium-dose XBB.1.5 RBD stimulated neutralizing antibodies against the currently circulating variant under monitoring (VUM) KP.2 variant. A study reported that antibodies obtained from the serum of monovalent XBB.1.5 RBD-vaccinated hamsters could robustly neutralize KP.2 [13]. However, immunization with low-dose XBB.1.5 RBD failed to elicit neutralizing effects against the KP.2 pseudovirus, which carries F456L and R346T mutations that enhance its transmissibility and immune evasion abilities [35]. Our XBB.1.5 RBD-based vaccine formulated with Al(OH)_3_:CpG induced neutralizing antibody titers against Delta, BA.2, XBB.1.5, and JN.1 (10–82-fold increase) compared with the control, demonstrating that RBD-based vaccines can induce effective protection against SARS-CoV-2 variants. The XBB.1.5 monovalent mRNA vaccines (Moderna and Pfizer-BioNTech) and the XBB.1.5 spike protein vaccine (Novavax) effectively confer protection against XBB and JN.1 subvariants [39,40,41]. In general, RBD vaccines are known for their favorable safety profile [42]. The cross-protection offered by XBB.1.5 RBD-based vaccines against the Delta and Omicron variants supports the WHO’s recommendation to update the vaccine composition.

Vaccines based on the XBB.1.5/Delta RBD and XBB.1.5/Beta RBD bivalent formulations have provide better broad-spectrum protection against various SARS-CoV-2 variants, including the Beta variant, when compared to the XBB.1.5 monovalent vaccine. However, the XBB.1.5/BA.2 bivalent vaccine did not show enhanced cross-protection over the XBB.1.5 monovalent vaccine. Sera from individuals immunized with XBB.1.5/Delta RBD and XBB.1.5/Beta RBD vaccines exhibited broad-spectrum protection against Beta, Delta, BA.2, XBB.1.5, XBB.1.16, JN.1, and KP.2 variants, as opposed to sera from the control group. Importantly, immunization with XBB.1.5/Delta and XBB.1.5/Beta stimulated neutralizing effects against the Beta pseudovirus, indicating that RBD-based bivalent vaccines offer extensive protection against SARS-CoV-2 variants. Previous research has shown that both Beta and Delta RBDs can elicit neutralizing responses against the SARS-CoV-2 Beta variant [21,22]. Bivalent vaccines may induce a more robust immune response against early variants that could potentially re-emerge [32], compared to monovalent vaccines.

Glycoengineered *P. pastoris* enables the production of the SARS-CoV-2 RBD. Our lab reported that the SARS-CoV-2 prototype vaccine and bivalent vaccine (expressed in glycoengineered *P. pastoris*) exhibited excellent immunogenicity and protection against SARS-CoV-2 in mice. In this study, we expressed the XBB.1.5 RBD in glycoengineered *P. pastoris* and analyzed its N-glycan composition. Our results revealed that mammalian A2G(4)2 glycans were the predominant N-glycans in the XBB.1.5 RBD. The N-glycosylation of RBD antigens is critical for antigen presentation, humoral immunogenicity, and immunoreactivity [7]. The XBB.1.5 RBD-based monovalent vaccines and XBB.1.5/Beta RBD, XBB.1.5/Delta RBD, and XBB.1.5/BA.2 RBD-based bivalent vaccines elicited high antibody titers in mice against SARS-CoV-2 variants. Moreover, the high-dose immunization groups showed no significant difference in inducing neutralizing antibodies against SARS-CoV-2 variants. Our results suggest that all XBB.1.5 RBD-based monovalent and bivalent vaccines, formulated with Al(OH)_3_:CpG, elicited humoral immunity in mice against SARS-CoV-2 variants.

The RBD is an attractive target for coronavirus vaccines [8]. Compared to vaccines based on the full-length spike protein (e.g., Novavax), RBD-based vaccines have smaller antigens, which reduces the generation of non-neutralizing antibodies and potential side effects. The majority of the antibodies induced by RBD vaccines exhibit neutralizing activity; however, only a few antibodies induced by spike protein vaccines exhibit neutralizing activity, and they may also target non-critical regions (e.g., NTD or S2) [42]. Additionally, glycoengineered yeast cells provide a platform for the rapid and large-scale production of RBD proteins, which is crucial for the production of candidate vaccines against SARS-CoV-2. In this study, we prepared monovalent and bivalent RBD-based vaccines using glycoengineered *P. pastoris*. Immunization with the XBB.1.5/Delta and XBB.1.5/Beta RBD-based bivalent vaccines enhanced the immunogenicity in mice against previously and currently circulating SARS-CoV-2 variants, indicating their potential use against current or future SARS-CoV-2 infections.

## 5. Conclusions

The glycoengineered *P. pastoris* was used in the production of the SARS-CoV-2 RBD with mammalian N-glycosylation, which enables the quick update of SARS-CoV-2 subunit vaccines. The XBB.1.5 RBD protein expressed in glycoengineered *P. pastoris* induced broad protection against the SARS-CoV-2 Delta, BA.2, XBB.1.5, XBB.1.16, JN.1, and KP.2 variants, but not Beta. Immunization with the XBB.1.5/Delta and XBB.1.5/Beta RBD-based bivalent vaccines conferred protection against currently and previously circulating SARS-CoV-2 subvariants, including Beta, in mice.

## Figures and Tables

**Figure 1 vaccines-13-00543-f001:**
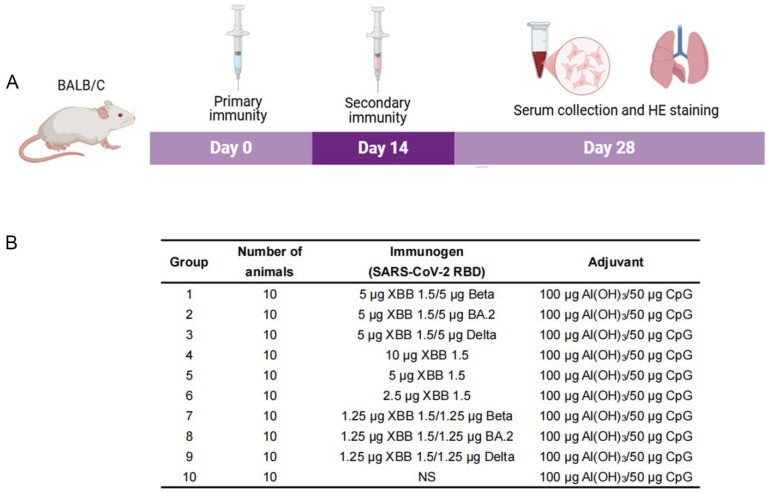
Timeline (**A**) and program (**B**) of mouse immunization.

**Figure 2 vaccines-13-00543-f002:**
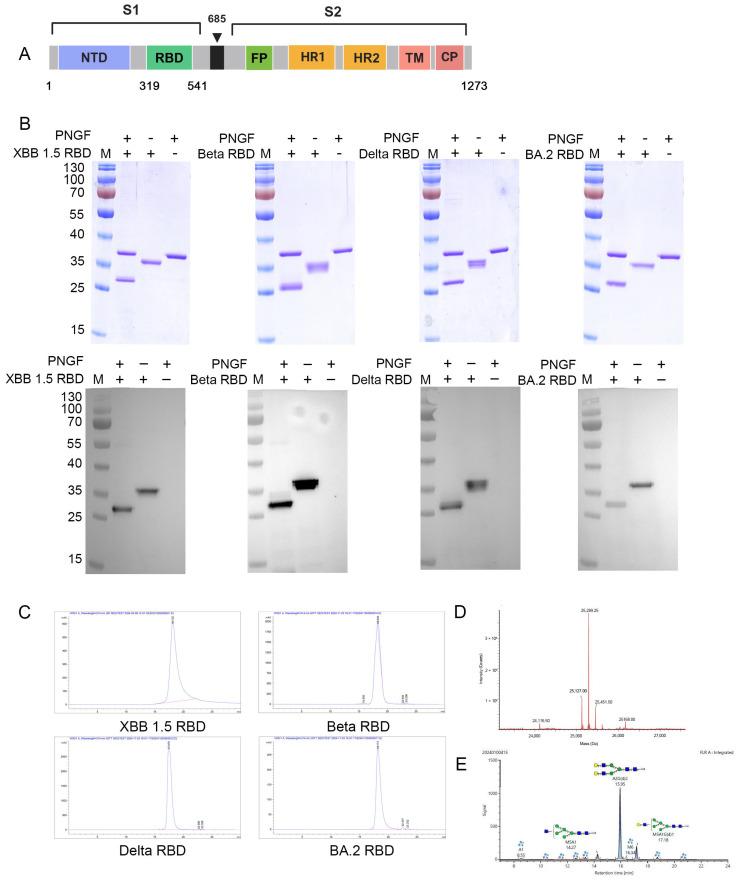
(**A**) Schematic diagram of the SARS-CoV-2 spike protein. NTD: N-terminal domain, RBD: receptor-binding domain, FP: fusion peptide, HR1: heptad repeat 1, HR2: heptad repeat 2, TM: transmembrane domain, and CP: cytoplasmic domain. (**B**) Mutations in the RBDs of different SARS-CoV-2 variants, including Beta, Delta, Omicron BA.2, and XBB.1.5. (**C**) SDS-PAGE and Western blot results of the Peptide-N-Glycosidase F (PNGase F)-digested RBD proteins of the SARS-CoV-2 XBB.1.5, Beta, Delta, and BA.2 variants. (**C**) Size-exclusion chromatography (SEC-HPLC) profiles of purified RBD proteins from SARS-CoV-2 XBB.1.5, Beta, Delta, and BA.2 variants. (**D**,**E**) The molecular weight (**D**) and N-glycoform (**E**) of the XBB.1.5 RBD protein were determined by ultra-performance liquid chromatography (UPLC) and mass spectrometry (MS). A: Antennary, G: Galactoses, M: Mannose.

**Figure 3 vaccines-13-00543-f003:**
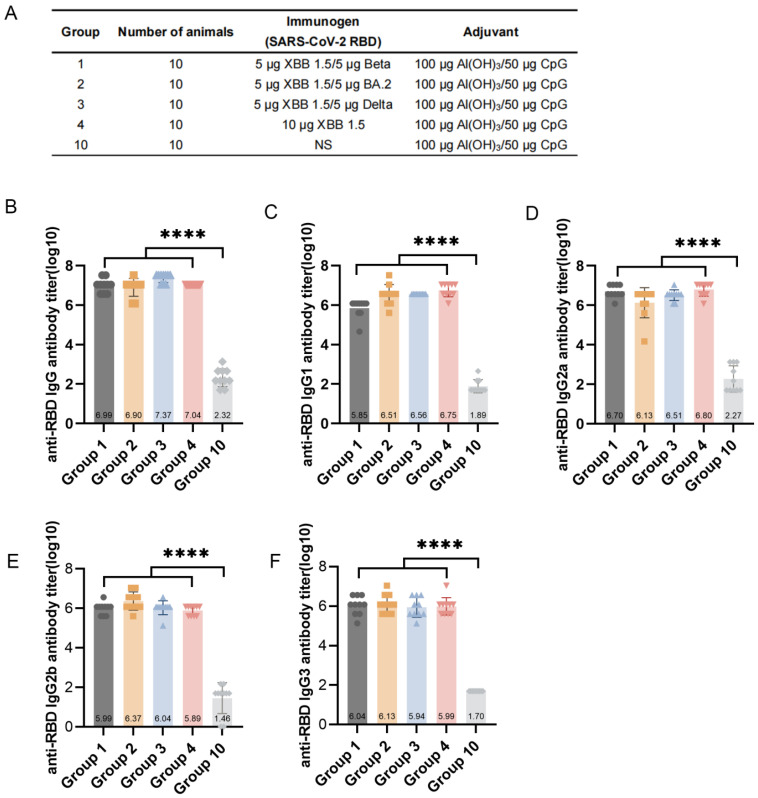
Humoral immune response in mice induced by receptor-binding domain (RBD)-based monovalent or bivalent vaccines. Mice (n = 10) were immunized with two doses of RBD-based monovalent or bivalent vaccines on day 0 and day 14. Serum samples were collected 28 days after the first vaccination. (**A**) Program of mouse immunization. (**B**–**F**) IgG (**B**), IgG1 (**C**), IgG2a (**D**), IgG2b (**E**), and IgG3 (**F**) titers of mice against SARS-CoV-2 RBD after vaccination with 5 µg XBB.1.5/5 µg Beta RBD, 5 µg XBB.1.5/5 µg BA.2 RBD, 5 µg XBB.1.5/5 µg Delta RBD, 10 µg XBB.1.5 RBD, and 100 µg Al(OH)_3_–50 µg CpG. Statistical significance was analyzed via *t*-tests; **** *p* < 0.0001.

**Figure 4 vaccines-13-00543-f004:**
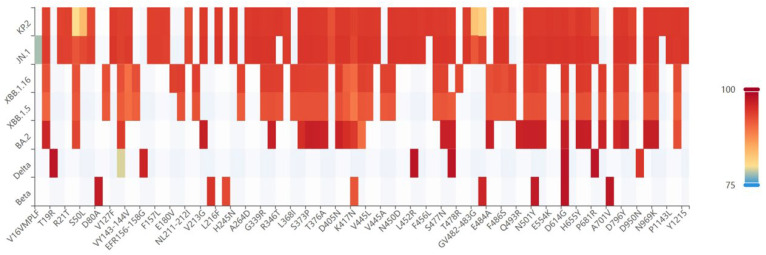
Mutation frequency heatmap of XBB.1.5 and other SARS-CoV-2 lineages based on data obtained from the National Genomics Data Center. The heatmap shows mutations with a frequency >0.75.

**Figure 5 vaccines-13-00543-f005:**
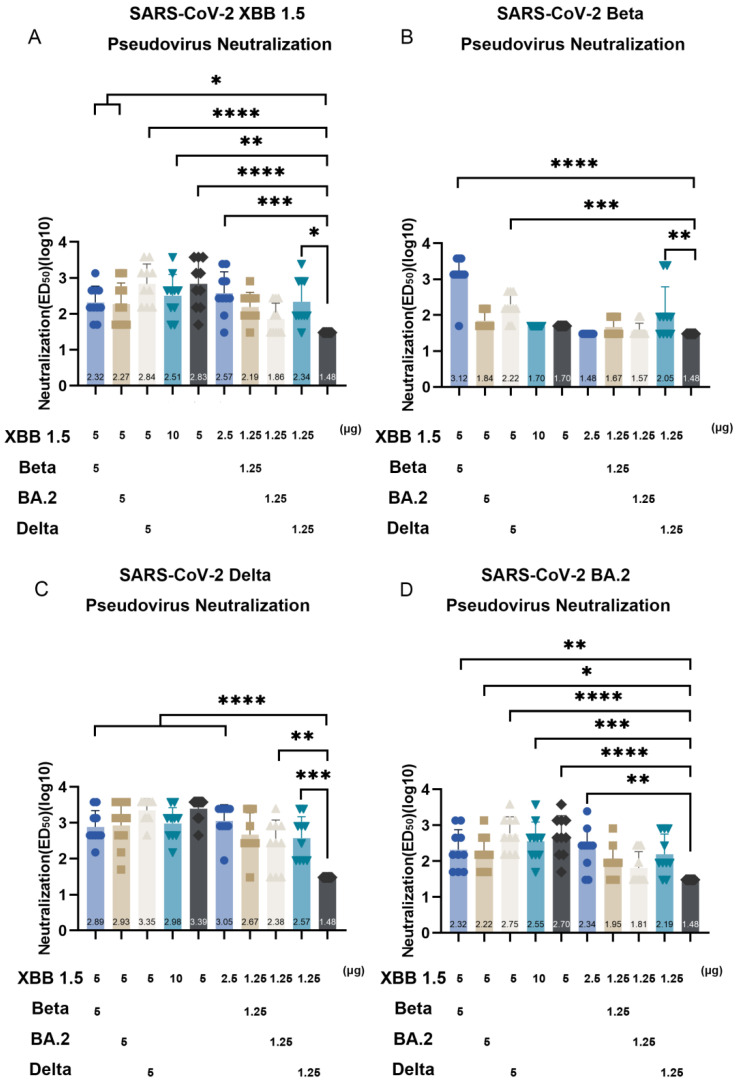
Neutralizing activity of sera from mice immunized with XBB.1.5 receptor-binding domain (RBD)-based vaccines against SARS-CoV-2 pseudoviruses XBB.1.5 (**A**), Beta (**B**), Delta (**C**), BA.2 (**D**), JN.1 (**E**), and KP.2 (**F**). (**G**) Neutralizing activity of sera from mice immunized with XBB.1.5 RBD-based vaccines against authentic SARS-CoV-2 XBB.1.16 variant. Mice (n = 10) received two intramuscular doses of SARS-CoV-2 RBD-based monovalent or bivalent vaccines on day 0 and day 14; serum was collected on day 28. The Y-axis shows the ED_50_ values of the neutralizing antibody titers on a log 10 scale, while the X-axis represents the monovalent or bivalent vaccine immunization group. Statistical significance was analyzed via *t*-tests; * *p* < 0.05; ** *p* < 0.01; *** *p* < 0.001; **** *p* < 0.0001.

**Table 1 vaccines-13-00543-t001:** Beta, Delta, Omicron BA.2, and XBB.1.5 RBD mutations relative to prototype.

WHO Label	Pango Lineage	RBD Mutation Sites
Beta	B.1.351	N501Y, E484K, K417N
Delta	B.1.617.2	T478K, L452R
Omicron	BA.2	G339D, S371F, S373P, S375F, T376A, D405N, R408S, K417N, N440K, G446S, S477N, T478K, E484A, Q493R, G496S, Q498R, N501Y, Y505H
Omicron	XBB 1.5	G339H, R346T.L368I, S371F, S373P, S375F, T376A, D405N, R408S, K417N, N440K, V445P, G446S, N460K, S477N, T478K, E484A, F486P, Q490S, G496S, Q498R, N501Y, Y505H

## Data Availability

Raw data supporting the conclusions of this paper will be provided by the authors without reservation.

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
