# Peer review of "XBB.1.5 RBD-Based Bivalent Vaccines Induced Antibody Responses Against SARS-CoV-2 Variants in Mice"

_vaccines, 2025, doi:10.3390/vaccines13050543_

Round 1
Reviewer 1 Report
Comments and Suggestions for Authors
Please see attached report

Please see attached report
Author Response
Dear reviewer,
Thank you for giving us the opportunity to revise our manuscript titled "XBB.1.5 RBD-based bivalent vaccines induced broad-spectrum protection against SARS-CoV-2 variants in mice". We appreciate the time and effort the reviewers dedicated to evaluating our work, and we have carefully addressed all their comments and suggestions. Below, we provide a point-by-point response to the reviewers' comments and detail the revisions made to the manuscript. Changes to the manuscript are highlighted in the revised version using a dark blue font.

Reviewer 2 Report
Comments and Suggestions for Authors
The authors studied the antibody responses to the vaccines generated based on SARS-CoV-2 variant XBB.1.5 spike protein RBD. The candidate vaccine antigens were formulated with adjuvant Al (OH)3:CpG and immunized mice monovalently or bivalently by intramuscular injection. Serum IgG antibody was then evaluated after two-doses immunization. Their results showed that monovalent XBB.1.5-RBD vaccine induced serum IgG and neutralizing antibodies against SARS-CoV-2 Delta, Omicron earlier variants, while bivalent vaccine containing XBB.1.5/Delta-RBD or XBB.1.5/Beta-RBD induced broader spectrum neutralizing antibodies in mice against Beta, Delta and Omicron variants including BA.2, XBB.1.5, XBB.1.16, JN.1, and KP.2. The topic of developing new vaccines are certainly of importance in preparation for timely control of infectious diseases in the future. However, the reported vaccine generation platform, antigen selection and adjuvant selection in this study are lack of novelty, the contribution of the study to the field is limited. Another shortcoming of this study is lacking in vivo virus challenging experiments, evaluation of the vaccine protective efficacy without this data was not complete. The manuscript was not well organized, some parts of the data were mixed with methods, and the figures are not presented in order.
Other minor points:
- During COVID-19 pandemic, natural infection and vaccination have built up human immunity against SARS-CoV-2 virus. In general, another COVID-19 pandemic in the future is unlikely. Please consider to revise the first sentence in the abstract.
- Even the authors and others have published studies using the same adjuvant combination. It is better for the reader understand the reason for the selection of these adjuvants combination and dosage in this mouse immunization study, some explanation for this would be better.
- As said above, the data (Figure 2 and Figure 4) should not be included in the Methods. They should be put in Results.
- Figure 1. As the mutation in were listed in figure 1B, animo acid numbering were needed in 1A, as least for the location of RBD.
- Figure 2: mouse gender, age, how many mice are used in this study are not clear. For some tests (determination of Tfh, GC B cells in lymphoid organ) N=3 is not adequate for statistic analysis.
- At 7 days after vaccination, mouse samples were taken for determination of Tfh, GC B cells, please clarify 7days after primary or boosting? what is the vaccine injection site, what lymph nodes were sampled? Please indicate what cell markers were used to define Tfh, GC B cells.
- Figure 2: As said before, most of the people has experienced SARS-CoV-2 infection or vaccination, whether the pre-existing immunity will affect the vaccination induced antibody responses? This question needs to be answered by study of mice with previous SARS-CoV-2 infection. –
- Figure2, IgG against RBD was determined, please indicate the RBD from which variants?
- What are advantages and disavantiges of your XBB.1.5 vaccines comparing to the XBB.1.5 monovalent mRNA vaccines by Moderna and Pfizer-BioNTech, and the XBB.1.5 spike protein vaccine by Novavax? This should be discussed.
- In Line 197, dosage for XBB.1.5 was missing.
Comments on the Quality of English Language
None
Author Response
Dear reviewer,
Thank you for giving us the opportunity to revise our manuscript titled "XBB.1.5 RBD-based bivalent vaccines induced broad-spectrum protection against SARS-CoV-2 variants in mice". We appreciate the time and effort the reviewers dedicated to evaluating our work, and we have carefully addressed all their comments and suggestions. Below, we provide a point-by-point response to the reviewers' comments and detail the revisions made to the manuscript. Changes to the manuscript are highlighted in the revised version using a dark blue font.
Comment 1:
The topic of developing new vaccines are certainly of importance in preparation for timely control of infectious diseases in the future. However, the reported vaccine generation platform, antigen selection and adjuvant selection in this study are lack of novelty, the contribution of the study to the field is limited.
Response 1:
Thank you for pointing this out.. We agree that next-generation platforms, including nanoparticle designs and novel adjuvants, are crucial for advancements in vaccinology. Nonetheless, our study deliberately concentrated on optimizing short-term solutions within current frameworks, which continue to serve as the foundation for global vaccine research and development. We are now expanding this research to assess these antigens in conjunction with more novel platforms, as proposed, to further improve immunogenicity.
The RBD-based vaccine formulated with (Al(OH)3:CpG) was selected because of their well-established safety, immunogenicity, and scalability in previous studies. Our aim was not to introduce a new platform but to evaluate whether updated antigen designs, informed by evolving SARS-CoV-2 variants could enhance broad-spectrum protection. Glycoengineered P. pastoris enable the quick and large-scale production of RBD with mammalian N-glycosylation, which enables the quick update of vaccine for SARS-CoV-2. Based on glycoengineered P. pastoris, the use of an established adjuvant and platform ensures that our findings are directly translatable to current vaccine manufacture, which prioritize proven safety profiles and rapid adaptability.
The key innovation lies in the antigen combination strategy. While monovalent XBB.1.5-RBD vaccines have been explored, our study systematically compared monovalent and bivalent formulations (XBB.1.5/Delta or Beta RBD) to address a critical question: does combining earlier variant antigens with newer Omicron subvariants improve broad protection? Our data revealed that bivalent vaccines induced significantly broader neutralizing antibodies against emerging variants like JN.1 and KP.2, which are now globally dominant and exhibit substantial immune evasion. This finding has directly inspired current debates regarding the utility of multivalent vaccines against rapid SARS-CoV-2 evolution..
Comment 2:
Another shortcoming of this study is lacking in vivo virus challenging experiments, evaluation of the vaccine protective efficacy without this data was not complete.
Response 2:
Thank you for pointing this out. The protective efficacy of the bivalent vaccine was not directly assessed through in vivo challenge experiments. Nonetheless, numerous studies have demonstrated a strong correlation between neutralizing antibody titers and protection efficacy in SARS-CoV-2 animal models or clinical trials.[1, 2]. Neutralization titer is an important predictor of vaccine efficacy in the future[3]. Therefore, in our study, neutralizing antibody titers assessed by pseudovirus neutralization test and live virus neutralization test are sufficient to verify the protective efficacy of the vaccine.
Comment 3:
The manuscript was not well organized, some parts of the data were mixed with methods, and the figures are not presented in order.
Response 3:
Thank you for point this out. We have redefined the insertion position and order of the figures to clearly separate the Methods and Results sections.
Comment 4:
During COVID-19 pandemic, natural infection and vaccination have built up human immunity against SARS-CoV-2 virus. In general, another COVID-19 pandemic in the future is unlikely. Please consider to revise the first sentence in the abstract.
Response 4:
Thank you for point this out. We agree that the emergence of another COVID-19 pandemic in the future is improbable. The first sentence in the abstract have be altered to "The currently circulating variant of Severe Acute Respiratory Syndrome Coronavirus-2 (SARS-CoV-2) exhibits resistance to antibodies induced by vaccines.."
Comment 5:
Even the authors and others have published studies using the same adjuvant combination. It is better for the reader understand the reason for the selection of these adjuvants combination and dosage in this mouse immunization study, some explanation for this would be better.
Response 5:
Thank you for point this out. We agree to clarify the rationale behind selecting the adjuvant combination (Al(OH)₃ + CpG) and its dosage in mice. We have added the following information to the introduction part of the manuscript: "Adjuvants aid in enhancing specific immune responses to the antigens present in vaccines. CpG2006 (CpG-7909), a synthetic oligodeoxynucleotide containing CpG motifs ('CpG ODN'), belongs to the class of TLR9 agonists, which can be recognized by TLR9 (Toll-like receptor 9)[27]. CpG effectively elicits the secretion of Th1-type cytokines, including interleukin (IL)-6, IL-12, and interferon gamma, through detection by TLR9 in B, T, and natural killer cells[28]. Aluminum hydroxide [Al(OH)3] is the most commonly used aluminum-based adjuvant, preferentially stimulating Th2-type immune responses by enhancing antigen persistence and recruiting innate immune cells[29]. In a previous study, bivalent vaccines, such as prototype/Beta RBD and Omicron/Delta RBD adjuvanted with 50µg CpG and 100µg Al(OH)3, were shown to induce a robust antibody response in mice and provide broad-spectrum protection.In previous studies, RBD adjuvanted with 50µg CpG of and 100µg of Al(OH)3 has been shown to induce a robust antibody response in mice and provide broad-spectrum protection "
Comment 6:
As said above, the data (Figure 2 and Figure 4) should not be included in the Methods. They should be put in Results.
Response 6:
Agree. We have put the data (Figure 2 and Figure 4) in Results.
Comment 7:
Figure 1. As the mutation in were listed in figure 1B, animo acid numbering were needed in 1A, as least for the location of RBD.
Response 7:
Agree. We have added the amino acid numbering of the location of RBD in Figure 1A.
Comment 8:
Figure 2: mouse gender, age, how many mice are used in this study are not clear.
Response 8:
Thank you for point this. Female mice,the information of mouse gender, age, the number of mice was added in the method part of the manuscript.
Comment 9:
For some tests (determination of Tfh, GC B cells in lymphoid organ) N=3 is not adequate for statistic analysis. At 7 days after vaccination, mouse samples were taken for determination of Tfh, GC B cells, please clarify 7days after primary or boosting? what is the vaccine injection site, what lymph nodes were sampled? Please indicate what cell markers were used to define Tfh, GC B cells.
Response 9:
Thank you for point this out.We observed an increase in the ratio of Tfh or GC B cells align with the results of neutralizing antibody titers and breadth, supporting the biological relevance of these findings. Crotty et al. utilized N=3 mice per group in to study Tfh cells and GC B cells change after bo.
Mice samples were taken for determination of Tfh, GC B cells 7days after primary vaccination. The mice were intramuscularly injected with vaccines. Mice inguinal lymph nodes were extracted.
T follicular helper (Tfh) cells are characterized by cell makers: CD4+CXCR5+PD-1+. Germinal center B cells are characterized by cell makers: CD45R+GL-7+CD95+. All these information was added in the Methods of the manuscript.
Comment 10:
Figure 2: As said before, most of the people has experienced SARS-CoV-2 infection or vaccination, whether the pre-existing immunity will affect the vaccination induced antibody responses? This question needs to be answered by study of mice with previous SARS-CoV-2 infection.
Response 10:
Thank you for pointing this out. We agree that studying mice with previous SARS-CoV-2 infections can elucidate whether pre-existing immunity impacts the antibody responses induced by vaccination. However, we are unable to conduct this experiment due to the constraints of a Biological Safety Protection Third-level Laboratory. Compared with full-length spike protein-based vaccines or inactivated vaccines, RBD-based vaccines have smaller antigens, which lower the generation of non-neutralizing antibodies, potentially avoiding the effect of pre-existing immunity. Recombinant RBD vaccines (e.g., ZF2001) remain effective in reducing viral loads in pre-infected mice, demonstrating their adaptability to pre-existing immunization[4]. Thus, our vaccine induces cross-protection against both early and current variants, suggesting it may circumvent pre-existing immune escape mechanisms due to mutation.
Review 11:
What are advantages and disavantiges of your XBB.1.5 vaccines comparing to the XBB.1.5 monovalent mRNA vaccines by Moderna and Pfizer-BioNTech, and the XBB.1.5 spike protein vaccine by Novavax? This should be discussed.
Response 11:
Thank you for highlighting this. Our bivalent RBD vaccines utilize a bivalent antigen presentation approach to enhance broad protection, setting them apart from monovalent mRNA or full-length S protein vaccines. By combining the XBB.1.5 RBD with the Delta/Beta RBD, bivalent RBD vaccines enhance immune responses against conserved epitopes, which may prevent immune escape compared to targeting a single strain. The RBD is the critical region of the SARS-CoV-2 spike protein responsible for binding to the human ACE2 receptor. Subunit vaccines, such as those targeting the RBD, minimize risks of vaccine-associated enhanced disease or systemic inflammation. Clinical trials of RBD vaccines reported lower rates of fever and fatigue compared to mRNA vaccines[5]. All this information was summarized in the Discussion section of the manuscript.
Thank you again for your consideration. Please do not hesitate to contact us if further
adjustments are required.
Sincerely
Bo Liu
Laboratory of Advanced Biotechnology, Beijing Institute of Biotechnology, Beijing,
China
liubo7095173@163.com
1. McMahan, K., et al., Correlates of protection against SARS-CoV-2 in rhesus macaques. Nature, 2021. 590(7847): p. 630-634.
2. Gilbert, P.B., et al., Immune correlates analysis of the mRNA-1273 COVID-19 vaccine efficacy clinical trial. Science, 2022. 375(6576): p. 43-50.
3. Khoury, D.S., et al., Neutralizing antibody levels are highly predictive of immune protection from symptomatic SARS-CoV-2 infection. Nature Medicine, 2021. 27(7): p. 1205-1211.
4. Qu, X., et al., Mouse model for pangolin-origin coronavirus GX/P2V/2017 infection and cross-protection from COVID-19 ZF2001 subunit vaccine. hLife, 2023. 1(1): p. 35-43.

Round 2
Reviewer 1 Report
Comments and Suggestions for Authors
Please see attached report

Please see attached report
Author Response
Dear reviewer,
Thank you for giving us the opportunity to revise our manuscript titled "XBB.1.5 RBD-based bivalent vaccines induced Antibody Responses against SARS-CoV-2 variants in mice". We appreciate the time and effort the reviewers dedicated to evaluating our work, and we have carefully addressed all their comments and suggestions. Below, we provide a point-by-point response to the reviewers' comments and detail the revisions made to the manuscript. Changes to the manuscript are highlighted in the revised version using a red font.
Comment 1: Page 1: It seems that all seven authors are acting as correspondents. Is this acceptable?
Response 1: Thank you for pointing this.This mistake has been corrected in the manuscript.
Comment 2:
The organization of the manuscript still needs attention. I am not convinced with the
authors’ response to my first comment on organization e.g., why is the control group
was number 5 not 1 or 10? Also, in figure 4, why the vaccination regimens are
shown in panel H not A?!! In addition, Figure labeling and references to these
figures is still confusing. For example the first figure in the article is Figure 1B
followed by figure 2A then Figure 4H!!! Figures must be numbered in the order of
appearance in the text. All these figure parts can be a single figure for the Materials
and Methods section and the remaining figures can be redesigned
Response 2: Thank you for pointing this out. All figures have been reorganized in the order of the appearance in the manuscript. The group number of control group has been changed to group 10. Some figures like program of immunization and RBD mutations has been removed to the Materials and method parts.
Comment 3:
My concern on the flow cytometric data definition of Tfh cells based on the technical
protocol shown in the Methods section in the absence of CD3 staining was not
satisfactory to me. The authors define Tfh cells by CD4+CXCR5+PD-1 +staining. The use of CD3 is essential and is much better than using CXCR5 and PD-1 that are expressed on other cells. The same applies for B cell definition by CD45R+GL-7+CD95+staining. These could be indicated as technical limitations!
Response 3: "We thank the reviewer for raising this critical point regarding the specificity of our flow cytometry gating strategies. We fully agree that CD3 is a canonical marker for T cell identification and that CXCR5/PD-1 can be expressed on non-T cell populations. Therefore, we have delete this part of results in the manuscripts.
Comment 4: Still, the links in lines 53, 62, 64, 66 and 72 in the new version are not working.
Response 4: Thank you for point this. We paste the link into the browser. The links on lines 48, 57, 61, and 66 are still operational.
Comment 5: Language of the mnuscript needs extensive edits as there are several word usage and style issues. For example,
- a) The comment regarding acronyms e.g., PBST, ED50, TCID50, VUM, PNGF…
that should be spelled out and abbreviated at their first appearance and used as
abbreviated later was not fixed.
- b) Line 48: there are two periods.
- c) Line 95: Vaccine RBD is joined as one word.
- d) Line 99-101: It is not clear what the authors want to convey in the sentence
starting with “These bivalent vaccines…). This sentence needs attention.
- e) Line 120: “.Anti-SARS-CoV”…. needs attention.
- f) Line 193: 3000; is joined with “Sigma” and there is an extra part of the bracket.
- g) Lines 239-240: “to assessed” should be “to assess” and a period is missing at the
end of line 239.
- h) “respectively” should usually be preceded by a comma as “…., respectively” as
in line 326 and elsewhere.
- i) Line 330-332: the --> The, Line 331: joined words and “1:886 however” needs
attention.
- j) Line 422: “However, they are less effective against Omicron BA.4 and BA.5
than earlier SARS-CoV-2 variants still confer protection against Omicron
subvariants”. This sentence needs rephrasing to be clear and convey the correct
message.
- k) Line 429: Two periods exist after ref. # 39.
- l) Line 453: “Bivalent --> bivalent
Response 5: Thank you for your thorough review and pointing these mistakes out.
- We have gone through the manuscript and modified acronyms.
- We have deleted one period.
- This mistake has been modified
- This sentence has been change to “The Omicron/Delta RBD-based bivalent vaccinestill confer protection against Omicron subvariants, however, the antibody-mediated ability against the recently emerged Omicron variants such as BA.4 and BA.5 reduced.”
- Anti-SARS-CoVhas been changed to Anti-SARS-CoV-2
- This mistake has been modified.
- These mistakes have been modified
- All “respectively” have been modified to precede by a comma。
- This mistake has been modified.
- This sentence has been changed to “However, they are less effective against the Omicron BA.4 and BA.5 subvariants compared to earlier SARS-CoV-2 variants, though they still provide protection against Omicron subvariants.”
- We have deleted one period.
- “Bivalent” has been changed to“bivalent”.
Thank you again for your consideration. Please do not hesitate to contact us if further
adjustments are required. Sincerely
Bo Liu
Laboratory of Advanced Biotechnology, Beijing Institute of Biotechnology, Beijing, China
liubo7095173@163.com

Reviewer 2 Report
Comments and Suggestions for Authors
The authors have addressed most of mine concerns and questions. Considering the authors may not have access to biosafety level three facilities, I accept their explanation for using data from serum neutralization tests against pseudo-, and, live-viruses to support the claims of the vaccines efficacy. However, the title for this manuscript has to be changed by removing “protection”, while specify only serum neutralizing antibody responses were studied.
Here are some minor comments:
- In the abstract, Line 30-31, sentence “Therefore, the monovalent….2023-2024” seems unnecessary, I suggest to remove.
- The last sentence of abstract sound strange, please check.
- Line 95, duplication of “bivalent vaccines”?
- In the methods, sampling of mouse organs after immunization as mentioned, it was also illustrated in figure 2, however, there is no related study in this manuscript.
None
Author Response
Dear reviewer,
Thank you for giving us the opportunity to revise our manuscript titled "XBB.1.5 RBD-based bivalent vaccines induced Antibody Responses against SARS-CoV-2 variants in mice". We appreciate the time and effort the reviewers dedicated to evaluating our work, and we have carefully addressed all their comments and suggestions. Below, we provide a point-by-point response to the reviewers' comments and detail the revisions made to the manuscript. Changes to the manuscript are highlighted in the revised version using a red font.
Comment 1: However, the title for this manuscript has to be changed by removing “protection”, while specify only serum neutralizing antibody responses were studied.
Response 1: Thank you for pointing this out. The title has been changed to “XBB.1.5 RBD-based bivalent vaccines induced Antibody Responses against SARS-CoV-2 variants in mice”.
Minor comment 1:In the abstract, Line 30-31, sentence“Therefore, the monovalent….2023-2024” seems unnecessary, I suggest to remove.
Response 1: Thank you for pointing this out. This sentence has been deleted.
Minor comment 2: The last sentence of abstract sound strange, please check.
Response 2: Thank you for pointing this. This sentence has been changed to “The XBB.1.5/Beta and XBB.1.5/Delta RBD-based bivalent vaccines are considered as potential candidates for broad-spectrum vaccines against SARS-CoV-2 variants.”
Minor comment 3: Line 95, duplication of “bivalent vaccines”?
Response 3: Thank you for pointing this. The second “bivalent vaccines” has been deleted.
Minor comment 3: In the methods, sampling of mouse organs after immunization as mentioned, it was also illustrated in figure 2, however, there is no related study in this manuscript.
Response 4: Thank you fro pointing this.This part of the result has been added to the supplementary file.
Thank you again for your consideration. Please do not hesitate to contact us if further
adjustments are required. Sincerely
Bo Liu
Laboratory of Advanced Biotechnology, Beijing Institute of Biotechnology, Beijing, China
liubo7095173@163.com

Round 3
Reviewer 1 Report
Comments and Suggestions for Authors
Links in the introduction are now working but links at Lines 62 and 69 open the same page!
Comments on the Quality of English LanguageThe manuscript should be checked for English for better clarity. For example, check the sentence on lines 94-97.
Author Response
Dear reviewer,
Thank you for giving us the opportunity to revise our manuscript titled "XBB.1.5 RBD-based
bivalent vaccines induced broad-spectrum protection against SARS-CoV-2 variants in mice". We
appreciate the time and effort the reviewers dedicated to evaluating our work, and we have
carefully addressed all their comments and suggestions. Below, we provide a point-by-point
response to the reviewers' comments and detail the revisions made to the manuscript. Changes to
the manuscript are highlighted in the revised version using a purple font.
Comment 1: Links in the introduction are now working but links at Lines 62 and 69 open the same page!
Response 1: Thank you for pointing this. We have modified the links at line 62 and 69.
Comment 2: The manuscript should be checked for English for better clarity. For example, check the sentence on lines 94-97.
Response 2: Thank you for pointing this. We have checked the manuscript and improve language.
Thank you again for your consideration. Please do not hesitate to contact us if further adjustments
are required.
Sincerely
Bo Liu
Laboratory of Advanced Biotechnology, Beijing Institute of Biotechnology, Beijing, China
liubo7095173@163.com